# Attitudes of Nursing Staff in Hospitals towards Restraint Use: A Cross-Sectional Study

**DOI:** 10.3390/ijerph19127144

**Published:** 2022-06-10

**Authors:** Silvia Thomann, Gesche Gleichner, Sabine Hahn, Sandra Zwakhalen

**Affiliations:** 1Applied Research & Development in Nursing, School of Health Professions, Bern University of Applied Sciences, Murtenstrasse 10, 3008 Bern, Switzerland; sabine.hahn@bfh.ch; 2Cardiovascular Center, Department of Cardiology, Bern University Hospital, University of Bern, 3010 Bern, Switzerland; gesche.gleichner@insel.ch; 3Department of Health Services Research, Care and Public Health Research Institute, Maastricht University, Duboisdomein 30, 6229 GT Maastricht, The Netherlands; s.zwakhalen@maastrichtuniversity.nl; 4Living Lab in Ageing and Long-Term Care, Maastricht University, 6200 MD Maastricht, The Netherlands

**Keywords:** attitude, hospitals, nursing, restraint

## Abstract

The attitude of nursing staff towards restraint use can be decisive for whether restraints are used. So far, nursing staff’s attitudes have been studied primarily in long-term and mental health care settings, while findings from somatic acute care hospital settings are largely lacking. Therefore, we aimed to investigate (a) the attitudes of hospital nursing staff towards restraint use, and (b) the construct validity and reliability of a measurement instrument for use in hospital settings that was developed and validated in long-term care settings (Maastricht Attitude Questionnaire (MAQ)). Using a cross-sectional design, the attitudes of 180 nursing staff towards restraint use were assessed. The data were analysed descriptively and by means of regression analysis and factor analysis. We found that nursing staff in hospitals have a neutral attitude towards restraint use and that the MAQ, with minor adaptations, can be used in hospital settings, although further testing is recommended. Neutral attitudes of nursing staff have also been observed in long-term and mental health care settings, where changing attitudes were found to be challenging. Interventions at the national level (e.g., legal regulations) and management level (e.g., providing alternatives and changing institutional culture) are suggested.

## 1. Introduction

Internationally, it is undisputed whether restraint use should be reduced as much as possible [1,2,3]. Restraints are ‘interventions that may infringe [on] a person’s human rights and freedom of movement, including observation, seclusion, manual restraint, mechanical restraint and rapid tranquillisation’ [4]. The use of restraints is an encroachment of basic human rights and has negative consequences for patients (e.g., increased risk of falls, delirium, strangulation, and re-traumatisation) and (in-)formal caregivers (e.g., distress) [5,6,7,8].

In inpatient settings, nursing staff play crucial roles in the decision-making process as well as in the application of restraint, as they are most intensively involved in patient care [7,9,10]. It is well known that the decision-making process for the use of restraint is influenced not only by contextual (e.g., availability of guidelines) and patient-related factors (e.g., aggressive behaviour) but also by the individual conditions of the staff [7,11,12]. An essential condition (in any) decision-making process is the attitude one adopts, as this attitude guides the appraisal of the situation and the selection of the given options in the situation [13]. Attitude is defined as ‘the stored evaluations of or feelings toward persons, objects, events, situations, routines, instructions, goals, positions, ideas, behaviours, and issues’ [13]. Attitude becomes particularly relevant in the decision-making process when there is little time and motivation to conduct an effortful, feature-based analysis of the situation. Given the high workload that nurses in the inpatient setting also describe as a contributing factor to restraint use [14,15,16], it becomes apparent that time is often scarce and, therefore, a decision based on attitude is more likely to be made. Indeed, whether nursing staff have a favourable or critical attitude toward restraint can influence its use. Thus, knowing and addressing the attitude of nursing staff may be an important contributor to restraint reduction [9].

To date, research on nursing staff’s attitudes towards restraint use in the inpatient setting has focused mainly on long-term and mental health care. In the long-term care field, the findings about nursing staff’s attitudes are inconsistent. Using qualitative approaches, negative feelings were expressed, while surveys with standardised questionnaires indicated a slightly favourable attitude of nursing staff towards restraint use. Furthermore, it was reported that attitudes have hardly changed in the past decades [17,18,19]. In the mental health care field, it was found that attitudes have tended to become slightly more critical over the past decades, although the change in attitude was not highly pronounced [9]. Rather, it has been shown that the view is changing from a therapeutic paradigm to a safety paradigm [20]. In the somatic acute care hospital (henceforth referred to as ‘hospital’) setting, little is known about the attitudes of the nursing staff. Some studies have been conducted in the intensive care area (e.g., [10,21,22,23]); however, these focused more on reasons for the use of restraints (e.g., using the Perception of Restraint Use Questionnaire), on knowledge and application practices (partly using questionnaires designed for the study), or on attitudes assessed only using qualitative methods. Therefore, the aim of the current study is to assess nursing staff’s attitudes in a hospital setting using a standardised questionnaire and to identify their associations with staff characteristics. Since, to our knowledge, no validated instrument has been used to assess attitudes towards restraint use in hospital settings, we also aimed to test the construct validity and reliability of an instrument validated in long-term care settings for use in hospital settings.

## 2. Materials and Methods

### 2.1. Study Design, Setting, and Sampling

Using a cross-sectional design, nursing staff in a department of a Swiss university hospital were surveyed regarding their attitude towards restraint use. The department operates with 146 patient beds distributed over seven ‘general’ inpatient, two outpatient, and three high-dependency care units. All nursing staff at all qualification levels, including those still in training, were eligible. There were no exclusion criteria. In order to test the construct validity and reliability of a questionnaire, 5–10 participants per item (question) of a scale are recommended [24]. The highest number of items is found in the attitude scale (see Section 2.2), which contains 22 items. Accordingly, we aimed for a sample size of 110–220 participants.

### 2.2. Instrument

The Maastricht Attitude Questionnaire (MAQ, German version) was used with the developers’ permission [25,26]. So far, the MAQ has been used solely in long-term care settings and has proven to be valid and reliable [27,28,29]. The MAQ includes socio-demographic information (age and gender), work-specific information (workplace, highest professional qualification, and work experience), and three scales dealing with the attitude and perception of nursing staff regarding the use of restraints in health care. We chose this questionnaire as it was the only one known to us at the time of measurement that had been translated into German to measure attitudes of nursing staff towards restraint use. In addition, it has been shown that the patient group most affected by restraint use in hospitals are older, care-dependent, and cognitively impaired [30]; thus, this is a patient group more closely resembling patients in long-term care than in psychiatry. There is another scale, the Physical Restraint Knowledge, Attitude, and Practice Scale, that has also already been used to measure the attitude of nurses in hospital (ICU) [23]. This scale is also based on a scale further developed in the long-term care setting. In this scale, however, attitude is only a subscale. Moreover, this scale was not available in German at the time our study was conducted.

The first scale of the MAQ assesses the general attitude towards restraint use with 22 items (see first table in Section 3.2). The internal consistency of the attitude scale is reflected by a Cronbach’s alpha of 0.81 [27]. This scale consists of three subscales:Consequences of restraint use for the patient (10 items, e.g., *Patients experience the use of physical restraints as safe*; Cronbach’s alpha = 0.71)Reasons for restraint use (8 items, e.g., *Restraints reduce the risk of serious injury to patients*; Cronbach’s alpha = 0.77)Appropriateness of restraint use (4 items, e.g., *If we use physical restraints it is always necessary*; Cronbach’s alpha = 0.58)

The items were answered on a Likert scale ranging from 1 (strongly disagree) to 5 (strongly agree). The interpretation of the results is based on the mean score of all 22 items (sum of all items divided by the number of items): the higher the score, the more favourable the attitude is towards the use of restraints.

In the second scale of the MAQ, the perceived degree of restrictiveness for the patient and, in the third scale of the MAQ, the extent of own discomfort with the use of the specific restraint were assessed (both 16 items, see Section 3.2). The answers were given on a Likert scale ranging from 1 (not restrictive/not discomforting) to 3 (highly restrictive/highly discomforting). The interpretation of the results is based on the mean score. Higher scores indicate a higher perception of the degree of restrictiveness for the patient, and higher scores indicate a greater degree of discomfort in using restraints for nursing staff.

As the MAQ was developed for long-term care settings, minor adjustments were made to the wording based on the setting and context. The word ‘resident’ was replaced with the word ‘patient’. Likewise, the word ‘hospital’ was used instead of ‘nursing home’. For the qualifications, the nomenclature typical of the educational qualifications in the field of nursing and care of the Swiss Health Observatory [31] were used. Otherwise, no changes were made to the content or to the number of items of the questionnaire. As the questionnaire was used for the first time in a hospital setting, the construct validity and reliability was tested (see Section 2.4). This resulted in a slight adaptation of the factor structure and, accordingly, in the calculation of the mean scores of each scale (see Section 3). In this study, the Attitude scale comprised 19 items (see Section 3.2), the Discomfort scale comprised 14 items, and the Restrictiveness scale comprised 10 items (for both, see Section 3.2). All results presented in this study (e.g., mean scores of the scales in Table 1) were derived from the adapted scales.

### 2.3. Data Collection

Data collection took place between October and December 2020. Information on the study along with a link to the online questionnaire (using the platform SoSci Survey) were sent to all nursing staff in the department via their employee email. To increase the response rate, a total of 3 reminder emails were sent (after 2, 4, and 6 weeks) to all eligible participants. To prevent missing data, mandatory fields were marked for the items of the scales but not for the socio-demographic and work-specific information.

### 2.4. Data Analysis

The software R 4.1.0 [32] was used to analyse the gathered data. By means of descriptive analyses (number and percentages with 95% confidence interval for nominal variables; mean and standard deviation, median and interquartile range, and range for ratio variables), the sample was described in terms of socio-demographic and work-specific characteristics as well as its attitude, perceived restrictiveness, and discomfort (R packages used: tableone [33] and compareGroups [34]). In addition, the correlation among the mean scores of the three scales were analysed using Pearson’s correlation coefficients. As the MAQ was used in a hospital setting for the first time, construct validity was tested by means of factor analyses. We initially intended to perform a confirmatory factor analysis for the scale on attitude, as it can be assumed that the population examined was similar to the population in which the questionnaire was validated. However, we found that the data did not fit the theoretical construct (e.g., one item was negatively correlated with the factor). In such a case, it is recommended to conduct an exploratory factor analysis (EFA) [35]. In addition, in previous scientific publications using the MAQ, only information about reliability (Cronbach’s alpha) of the Attitude scale was published. Information on construct validity is lacking. For the two other scales (Discomfort and Restrictiveness), no statistical parameters based on factor analysis could be identified. Therefore, an EFA was carried out for all three scales of the MAQ, starting with the original number of items per scale (Attitude *n* items = 22; Discomfort *n* items = 16; Restrictiveness *n* items = 16).

For the EFA, the following analyses and cut-off values were used [36,37]: we identified factorability computing the correlation matrix, the Bartlett test of sphericity (*p*-value < 0.05), and the Kaiser–Meyer–Olkin (KMO) criterion (>0.5). In addition, we checked the measure of sampling adequacy (MSA) for each item (>0.5). To determine the number of factors to retain, we used several approaches: we interpreted the scree plot and used parallel analysis and the minimum average partials (MAP). We started the analysis with the highest recommended number of factors. If a factor had <3 items or if the allocation of the items to the factors did not make sense in terms of content, the next-smallest number of factors was trialled, up to the final number of factors. For the factor analysis, we used the “oblimin” rotation method, as earlier analyses of the attitude scale of the MAQ showed that these factors are correlated [29]. For item-factor loading, a cut-off value of >0.3 was used. In the case of an exclusion of items due to too-low factor loading, the above steps were repeated. Finally, internal consistency/reliability for each factor as well as for the full scale was calculated by means of the Cronbach’s alpha. Cases with missing values in one item of the scale were excluded. The results of the EFA on the Attitude scale were additionally compared with the original version. The R packages used were psych [38], corrplot [39], dplyr [40], and GPArotation [41].

Multiple linear regression analyses were carried out to identify associations between each of the three scale mean scores and staff characteristics. Attitude, discomfort, and restrictiveness were defined as the dependent variables, and the staff characteristics listed in Table 1 were defined as independent variables. Since years of work experience and age were correlated, only work experience was included in the model. Cases with missing values were excluded. The R packages used were MASS [42], tidyverse [43], and jtools [44].

As the professional qualification was highly heterogeneous, with very few answers per qualification in some cases, these had to be grouped for a meaningful analysis. This grouping was also based on the nomenclature typical of the educational qualifications in the fields of nursing and care of the Swiss Health Observatory [31]. For our analyses, the following four categories were used:RN BSc/MSc: Registered nurse (RN) with a Bachelor of Science (BSc) or Master of Science (MSc) in nursing;RN+: RN with a degree from a college of higher education (so-called Advanced Federal Diploma of Higher Education in Nursing; European Qualifications Framework: Level 6 [45]), and further education as an intensive care, anaesthesia, or emergency care nurse;RN: RN with a degree from a college of higher education (so-called Advanced Federal Diploma of Higher Education in Nursing; European Qualifications Framework: Level 6 [45]);Non-RN: Staff working in the field of nursing but not having an RN qualification (including 3-year vocational training in nursing (European Qualifications Framework: Level 4 [45]); staff with other degrees in the field, such as nursing assistants; students; trainees; and staff with other professional degrees outside the nursing field).

### 2.5. Ethical Considerations

The ethics committee assessed the project as not being subject to the Swiss Human Research Act (BASEC-Nr: Req-2020-01204). Participation in the survey was voluntary and anonymous. In order to participate in the survey, the participants provided informed consent before the start of the survey, as this was the first question. The survey could be stopped at any time without giving reasons.

## 3. Results

### 3.1. Sample and Attitude

A total of 351 nursing staff were invited to participate in the survey. Of these, 182 completed the survey, including all items of the general Attitude scale. Two participants gave implausible information about their age (−1 and 1); hence, these cases were excluded. Thus, 180 questionnaires could be included in the analysis, corresponding to a participation rate of 51.3%. There were further missing responses in both the Discomfort (*n* = 9) and Restrictiveness scales (*n* = 2) (Figure 1).

The majority of participants were female (91.7%), worked in an inpatient unit (51.7%) and were registered nurses with a qualification from a college of higher education (RN 48.9%; see Table 1). The median age of the participants was 35 years, and the median number of years of work experience was 13 years. The participants tend to have a neutral general attitude towards restraint use (mean 3.2 on a scale of 1–5) and to perceive the restrictiveness of restraints (mean 2.1 on a scale of 1–3) and the discomfort in their use (mean 2.2 on a scale of 1–3) as being moderate. However, the discomfort as well as the perceived restrictiveness differ greatly depending on the restraint type (see second table in Section 3.2). The sensor alarm was perceived by the participants as being both the least uncomfortable (mean 1.2) and least restrictive restraint (mean 1.4). The abdominal belt in bed was perceived to be the most discomforting (mean 3.0), and the ankle belt was perceived to be the most restrictive restraint (mean 2.9). Furthermore, the mean scores of the three scales are correlated: the greater the discomfort in the use of restraints, the more critical the attitude towards restraints (r = −0.22; *p* = 0.003); the stronger the perceived restrictiveness of restraints, the more critical the attitude towards restraints (r = −0.28, *p* = 0.000); and the greater the discomfort in the use of restraints, the more restrictive they are perceived (r = 0.52; *p* ≤ 0.000).

The linear regression analysis showed that the general attitude is associated with work experience. With increasing work experience, a more restraint-favouring attitude is taken (β 0.01). Other staff characteristics did not show a statistically significant association with the general attitude. The perceived discomfort in using restraints is associated with the staff’s workplace. Nursing staff who work in the high-dependency care unit feel less discomfort with the use of restraints than nursing staff in ‘general’ inpatient units (β −0.17). All other associations were not statistically significant. The perceived restrictiveness of restraints is associated with work experience, workplace, and professional qualification. Increasing work experience is associated with a lower perceived restriction of restraints (β −0.00). Nursing staff who work in the high-dependency care unit perceived restraints as being less restrictive than nursing staff in a ‘general’ inpatient unit (β −0.09). RN+ (β 0.20) or RN (β 0.11)-qualified staff tend to perceive restraints as being more restrictive than non-RN staff. Our models explain between 7% and 9% of the variance (R^2^; see Table 2).

### 3.2. Construct Validity and Reliability of the MAQ

For the general Attitude scale, we found that the factor structure of the original scale is largely similar to that in the hospital setting, see Table 3. Four items showed deviating results. Items 10 (*I always question why a restraint is applied on a patient* (*recoded*)) and 21 (*I would rather risk falling than be physically restrained in a chair all day* (*recoded*)) did not load sufficiently on any factor and were, therefore, excluded. Item 15 (*The adverse effects of physical restraints do not outweigh the increase in safety*) was negatively correlated with the factor and was, therefore, excluded. Item 13 (*Applying physical restraints usually has a calming effect on patients*) was loaded on Factor 2 (Reasons for restraint use) instead of Factor 1 (Consequences of restraint use for the patient), as in the original scale. Item 2 (*If we use physical restraints it is always necessary*) additionally showed cross-loadings. Here, the allocation to the factors of the original scale was considered appropriate from a content point of view. We also considered the factor naming to be appropriate (Factor 1 = Consequences of restraint use for the patient, α = 0.83; Factor 2 = Reasons for restraint use, α = 0.77; and factor 3 = Appropriateness of restraint use, α = 0.55). The adapted scale comprising 19 items explains 37% of the variance and has an internal consistency of α = 0.83.

For the Discomfort and Restrictiveness scales, we found that both scales contain two factors (see Table 4). However, some items were loaded below the required value (0.3) and were, therefore, excluded. For the Discomfort scale, this applies to items 14 and 15, and for the Restrictiveness scale, this applies to items 6, 7, 9, 11, 14, and 15. For the Discomfort scale, items 2, 4, and 11 were loaded on multiple factors with >0.3. Assignment to a factor was based on content. The two factors were named as follows: Fixation belts in bed (Factor 1) (items 10, 13, and 16; α = 0.90), and Mechanical and electronic restraint except fixation belts (Factor 2) (items 1–9, 11, and 12; α = 0.78). The Discomfort scale explained 38% of the variance in perceived discomfort and had an internal consistency of α = 0.78. For the Restrictiveness scale, there were no cross-loadings. The two factors were named as follows: Restraining the patient to the bed (Factor 1) (items 8, 10, 13, and 16; α = 0.66) and Safety measures in the chair when leaving the bed or place (Factor 2) (items 1–5 and 12; α = 0.63). The Restrictiveness scale explained 35% of the variance in perceived restrictiveness and shows an internal consistency of α = 0.65.

## 4. Discussion

In this cross-sectional study, we investigated the attitudes of nursing staff in hospitals towards restraints and the association of attitudes with staff characteristics. Based on data gathered from 180 participants, we found that nursing staff have a rather neutral attitude towards restraints in general and perceive the discomfort in the use and restrictiveness of restraints as being moderate. These three constructs are, furthermore, correlated as expected: the greater the discomfort or the stronger the perceived restrictiveness, the more critical the attitude towards restraint use, and the greater the discomfort, the more restrictive the restraints are perceived to be. The following associations between attitude/discomfort/restrictiveness and staff characteristics were found: general attitude and work experience; discomfort and working in the high-dependency care unit; and restrictiveness and working in the high dependency care unit, work experience, and qualification. In addition, we tested the construct validity and reliability of the MAQ for its use in hospital settings. We found that, with minor adaptations, the MAQ can also be used in hospital settings, although further testing is necessary.

### 4.1. The Attitudes of Nursing Staff

A neutral attitude of nursing staff towards restraint use has also been observed in studies using questionnaires in long-term care and mental health care settings. By using qualitative methods, more critical attitudes were identified [9,17,20,27]. With regard to associations, there are no consistent findings so far. Our model also explained little of the variance, and we only identified professional experience as being positively associated with the general attitude, i.e., with increasing professional experience, an attitude slightly favouring restraint use evolves. It is known that routine and institutional culture play important roles in restraint use in hospitals [16,17,46,47,48]. It is possible that with more professional experience, the prevailing routines will become internalised and the practice will be less critically scrutinised. However, the association is not pronounced and should be further investigated.

Both the discomfort in the use of restraints and the perceived restrictiveness show that nursing staff working in a high-dependency care unit perceive both to be less pronounced, compared with nursing staff in ‘general’ inpatient units. This is possibly related to a habituation effect, since in these units, as the name indicates, more complex patient situations are cared for, which has been shown to be related to the use of restraints [30]. The restrictiveness is also perceived to be less pronounced when nursing staff have more work experience. This, in turn, could be due to similar effects as with the general attitude, i.e., one questions the practice less critically and legitimises the use of restraints for oneself as a kind of coping strategy against distress that may occur when using restraints [7]. With regard to qualification, it can be seen that staff with an RN or RN+ degree perceive restraints as more restrictive than non-RN staff. One explanation may be found in the requirements for the different qualification levels. According to the European Qualifications Framework [45], non-RN staff (level 4 and below) are responsible for predictable situations and perform their work according to predefined guidelines. In contrast, RN staff (level 6) are responsible for complex, unpredictable situations and must be able to make decisions in these situations. In addition, at this qualification level and above, a critical reflection on theories and practices is expected. However, no significant difference in the perception of restraints could be identified between those with an RN BSc/MSc degree and those with non-RN degrees, which limits this interpretation. It is possible that the merger of the various qualification degrees into groups plays a role here.

With regard to the perception of discomfort in the use and the restrictiveness of the individual restraint types, the results are in line with previous findings in long-term care settings using the same questionnaire [27,29]. In general, it can be summarised that the more obvious the movement restriction, the more uncomfortable its use and the more restrictive the restraint is perceived to be. From our point of view, however, the results also imply that it is primarily the restriction of movement that is perceived, and the other forms of restriction of freedom are perceived less. On the one hand, it has been pointed out in previous studies that not all restraints are recognised as such because, among other reasons, clear and/or broader definitions of restraints seem to be little established in practice [12,17]. On the other hand, in many countries, it is primarily the restriction of movement that is regulated by law. In Switzerland, for example, the law only clearly regulates the restriction of movement for persons with compulsory admission and for persons who lack judgmental capacity and live in a care facility [49]. Thus, it seems important not only to raise awareness about the different forms of restraints but also to have clear legal regulations [18].

In the long-term care and mental health care settings, where restraint use has been researched for a longer time, it is evident that the attitude has hardly changed over the past decades [9,17]. Nursing staff’s attitudes tend to be neutral. However, given the growing evidence that the benefits of restraint do not outweigh the harms and the ethical guidelines that have been in place for some years, one would expect attitudes to become increasingly critical. That this does not happen could, in our view, be related to the following two aspects in relation to the concept of attitude [13]. First, it is described that evidence that does not correspond to one’s own attitude is often rejected or discounted. Second, restraint use is a routine practice [12,17] and routine is accompanied by a favourable attitude towards the practice. Nevertheless, it is also important to note that decision-making also means weighing different options. With regard to restraint use, it is known that alternatives are not very common or known [17,50]. The lack (of awareness) of these alternative options in the decision-making process may be another reason why the attitude of nursing staff towards restraint use is neutral and hardly changes. Therefore, it seems important to develop alternatives on the one hand and to change the perception of restraint use as a routine intervention or as being part of the job on the other hand. Indeed, changing attitudes proves that challenging, and neutral or even favourable attitudes toward restraint use also pose a barrier to the successful implementation of restraint reduction programs [18,51]. Therefore, it seems that both management and policy makers are required to promote the process of change with appropriate measures [52]. Management can ensure that options are available (alternatives/prevention options) or that they are not available (restraint material) in the decision-making process. In addition, management can play an active role in shaping the institutional culture of restraint use. Policy makers can further promote or stimulate these processes by providing clear and binding regulations and by monitoring their implementation in practice. In this way, it may be possible to change nursing staff’s attitudes towards restraint use over time.

### 4.2. Construct Validity and Reliability of the MAQ

Regarding the construct validity and reliability of the Attitude scale for its use in hospital settings, we found similar results to those previously reported [27]. Differences were primarily found for Factor 1 (Consequences of restraint use for the patient): two items were removed (item 15 and item 10) and one item (item 13) changed to Factor 2 (Reasons for restraint use), resulting in a higher reliability compared with the original scale (α = 0.83 vs. 0.71). The Discomfort and Restrictiveness scales have so far been analysed and interpreted mostly at the single item level [27,28,29]. Two of the studies also calculated a mean score of all items [27,28]. However, information on a possible factor structure or internal consistency is missing. Our factor analysis shows that there is a two-factor solution for both scales. The internal consistency (α) of the Discomfort scale is 0.78, and that of the Restrictiveness scale is 0.65. Two restraint types were not considered in the factor structure of either scale (*Bedroom door locked* and *Ward door locked*). These restraint types are unlikely to be used in hospital settings, which might explain the result.

While the scales were only tested for construct validity and reliability in this study, it seems advisable to further develop them. As the data from long-term care and mental health care settings show, changing attitudes towards restraint use seems to be challenging. The measurement of attitude could, therefore, be of use for raising awareness as well as for training purposes, or as a secondary outcome in intervention studies to reduce restraint use. From a content point of view, it should be reviewed whether the scale should be extended/adapted even further to other forms of restraint (e.g., pharmacological) or even extended to the broader concept of involuntary treatment. From a psychometric point of view, it is recommended to use a large sample size for further development and validation of the MAQ since the item-factor loadings are rather low [53,54]. Moreover, the scales only explain between 35% and 38% of the variance, which also indicates that further development of the scales is suggested. Aside from the instrument-related limitations, it is important to note the rather low participation rate, which limits the generalisability of the results.

## 5. Conclusions

Nursing staff in hospitals have a rather neutral attitude towards restraint use, as has already been found in mental health care and long-term care settings. Although the use of restraint is being critically scrutinised internationally and corresponding ethical guidelines have been developed, hardly any change could be observed in the long-term care setting and only a slight change in attitude has occurred in the mental health care setting over the past several decades. Policy makers and management are obliged to establish conditions that favour a change in attitude. In addition, the further development of instruments for the valid measurement of attitudes towards restraint use is recommended. This could help to monitor whether restraint-reduction initiatives also reach attitudes as an important component in the decision-making process. Such instruments may also be useful in education in order to raise awareness or to take appropriate initiatives early on so that nursing staff develop a critical attitude toward restraint use from the very beginning of their career.

## Figures and Tables

**Figure 1 ijerph-19-07144-f001:**
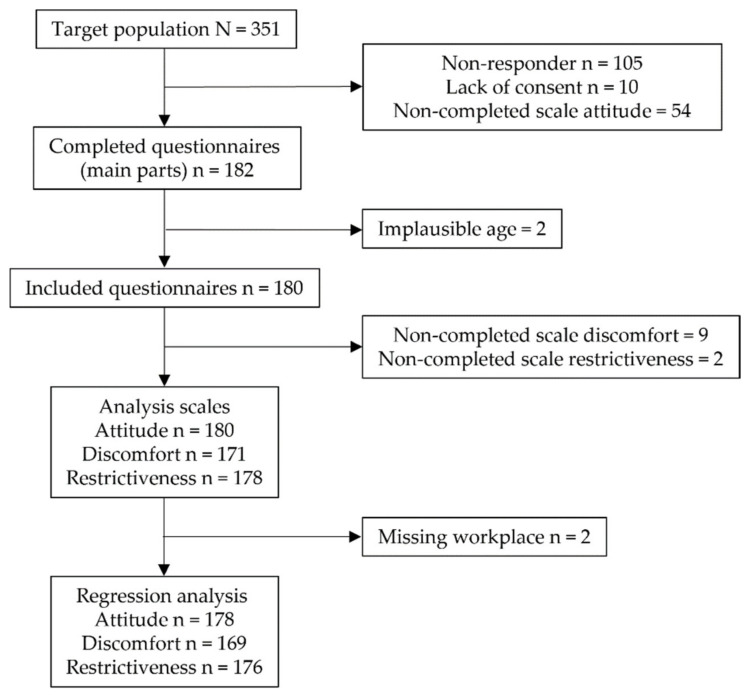
Flowchart detailing the number of participants per analysis step and scale.

**Table 1 ijerph-19-07144-t001:** Sample description.

Characteristics (*n* Answers)	*n* (% [95% CI])		
Sex (180)			
Female	165 (91.7 [86.6–95.3])		
Male	15 (8.3 [4.7–13.4])		
Workplace (180)			
Inpatient unit (*excluding the high-dependency care unit*)	93 (51.7 [44.1–59.2])		
Outpatient unit	39 (21.7 [15.9–28.4])		
High-dependency care unit	46 (25.6 [19.4–32.6])		
No response	2 (1.1 [0.1–4.0])		
Professional qualification (180)			
RN BSc/MSc	29 (16.1 [11.1–22.3])		
RN+	24 (13.3 [8.7–19.2])		
RN	88 (48.9 [41.4–56.4])		
Non-RN	39 (21.7 [15.9–28.4])		
	**Mean (SD)**	**Median (IQR)**	**Range**
Age in years (178)	36.7 (12.8)	35 (26–46)	16–69
Work experience in years (180)	16.0 (12.0)	13 (6–25)	0–45
Attitude (180)	3.2 (0.5)	3.2 (2.9–3.5)	1.5–4.6
Restrictiveness (178)	2.1 (0.2)	2.1 (1.9–2.3)	1.5–2.5
Discomfort (171)	2.2 (0.3)	2.2 (2.0–2.4)	1.4–2.8

*n* = number; 95% CI = 95% confidence interval; SD = standard deviation; IQR = interquartile range; RN BSc/MSc = Registered nurse with a Bachelor of Science (BSc) or Master of Science (MSc) in nursing; RN+ = Registered nurse with a degree from a college of higher education + further education as an intensive care, anaesthesia, or emergency care nurse; RN = Registered nurse with a degree from a college of higher education. Non-RN = staff with 3-year vocational training in nursing; staff with other degrees in the field, such as nursing assistants; students; trainees; or staff with other professional degrees outside the nursing field.

**Table 2 ijerph-19-07144-t002:** Associations between participants’ characteristics and their general attitude towards restraints, their discomfort in using restraints, and their perceived restrictiveness of restraints.

Predictor	Attitude (*n* = 178) *F*(7, 170) = 2.31, *p* = 0.028 R^2^/R^2^ Adjusted 0.09/0.05				Discomfort (*n* = 169) *F*(7, 161) = 1.65, *p* = 0.126 R^2^/R^2^ Adjusted 0.07/0.03				Restrictiveness (*n* = 176) *F*(7, 168) = 1.97, *p* = 0.037 R^2^/R^2^ Adjusted 0.08/0.04			
	β	SE	95% CI	*p*	β	SE	95% CI	*p*	β	SE	95% CI	*p*
(Intercept)	3.04	0.08	2.88–3.19	**<0.001**	2.23	0.05	2.13–2.33	**<0.001**	2.09	0.04	2.01–2.17	**<0.001**
Sex male	0.08	0.12	−0.17–0.33	0.525	−0.06	0.08	−0.21–0.10	0.459	0.03	0.06	−0.09–0.16	0.591
Work experience in years	0.01	0.00	0.00–0.02	**0.003**	−0.00	0.00	−0.01–0.00	0.305	−0.00	0.00	−0.01–0.00	**0.011**
Workplace												
Inpatient unit	Reference				Reference				Reference			
Outpatient unit	−0.04	0.09	−0.21–0.13	0.628	−0.06	0.06	−0.17–0.05	0.281	−0.01	0.04	−0.09–0.08	0.877
High-dependency care unit	−0.10	0.08	−0.27–0.07	0.230	−0.17	0.05	−0.27–0.06	**0.002**	−0.09	0.04	−0.18–0.01	**0.033**
Qualification												
Non-RN	Reference				Reference				Reference			
RN BSc/MSc	0.16	0.11	−0.06–0.38	0.141	0.01	0.07	−0.13–0.14	0.933	0.09	0.06	−0.03–0.20	0.131
RN+	−0.13	0.14	−0.41–0.14	0.337	0.14	0.09	−0.03–0.31	0.114	0.20	0.07	0.06–0.34	**0.004**
RN	0.05	0.09	−0.13–0.23	0.575	0.05	0.06	−0.07–0.16	0.450	0.11	0.05	0.01–0.20	**0.026**

*n* = number; *F* = F statistics; *p* = *p*-value (bold if significant); β = coefficients, SE = standard error, 95% CI = 95% confidence interval; SD = standard deviation; IQR = interquartile range RN BSc/MSc = Registered nurse with a Bachelor of Science (BSc) or Master of Science (MSc) in nursing; RN+ = Registered nurse with a degree from a college of higher education + further education as an intensive care, anaesthesia, or emergency care nurse; RN = Registered nurse with a degree from a college of higher education. Non-RN = staff with a 3-year vocational training in nursing; staff with other degrees in the field, such as nursing assistants; students; trainees; and staff with other professional degrees outside the nursing field.

**Table 3 ijerph-19-07144-t003:** Descriptive and factor analysis for the Attitude scale.

Attitude Bartlett’s χ^2^ = 86.98, df = 18, *p*-Value < 0.000 KMO 0.84 α Full Scale (95% CI): 0.83 (0.79–0.86) Explained Variance 37% (F1 16%; F2 15%; F3 6%)							
Item nr.	Label	Mean (SD)	Median (IQR)	Factor Original Scale	F1 (α 0.83 [95% CI 0.79–0.87])	F2 (α 0.77 [95% CI 0.71–0.82])	F3 (α 0.55 [95% CI 0.45–0.66])
					factor loading (α if item is dropped)		
01	My ward/unit uses physical restraints far too often (recoded)	4.3 (0.8)	4.0 (3.0–5.0)	Appropriateness			0.51 (0.49)
02	If we use physical restraints it is always necessary	4.3 (0.9)	4.0 (4.0–5.0)	Appropriateness		0.35	0.35 (0.49)
03	Physical restraints are used too quickly (recoded)	4.1 (0.9)	4.0 (4.0–5.0)	Appropriateness			0.62 (0.43)
09	Physical restraints are applied as a result of convenience of nursing staff (recoded)	4.4 (0.8)	5.0 (4.0–5.0)	Appropriateness			0.36 (0.50)
04	I’m afraid of falls if I do not apply physical restraints	2.7 (1.0)	3.0 (2.0–3.0)	Reasons		0.60 (0.73)	
05	It’s better to tie up patients than risk accidents	2.2 (1.0)	2.0 (1.0–3.0)	Reasons		0.47 (0.75)	
06	Falls in older adults often cause serious injury	3.6 (0.8)	4.0 (3.0–4.0)	Reasons		0.40 (0.77)	
07	Restraints reduce the risk of serious injury to patients	3.3 (0.9)	3.0 (3.0–4.0)	Reasons		0.66 (0.72)	
08	Failure to restrain puts individuals and facilities at risk for legal liability	2.9 (1.0)	3.0 (2.0–4.0)	Reasons		0.61 (0.73)	
11	Restraint-free care is impossible	2.3 (1.1)	2.0 (1.0–3.0)	Reasons		0.51 (0.76)	
12	The moral duty to protect people from harm requires restraint	2.8 (1.0)	3.0 (2.0–3.0)	Reasons		0.77 (0.70)	
21	I would rather risk falling than be physically restrained in a chair all day (recoded)	2.8 (1.2)	3.0 (2.0–4.0)	Reasons			
13	Applying physical restraints usually has a calming effect on patients	2.1 (0.8)	2.0 (2.0–3.0)	Consequences		0.32 (0.76)	
10	I always question why a restraint is applied on a patient (recoded)	3.0 (1.5)	3.0 (2.0–4.25)	Consequences			
15	The adverse effects of physical restraints do not outweigh the increase in safety	2.8 (0.9)	3.0 (2.0–3.0)	Consequences			
14	Applying physical restraints is a major cause of pressure ulcers (recoded)	3.6 (0.9)	4.0 (3.0–4.0)	Consequences	0.41 (0.82)		
16	Most patients suffer adverse effects from physical restraints (recoded)	3.4 (0.9)	3.0 (3.0–4.0)	Consequences	0.70 (0.80)		
17	Physical restraints reduce a patient’s quality of life (recoded)	2.8 (1.0)	3.0 (2.0–4.0)	Consequences	0.68 (0.79)		
18	Patients experience the use of physical restraints as a form of punishment (recoded)	2.9 (1.0)	3.0 (2.0–4.0)	Consequences	0.88 (0.79)		
19	Patients experience the use of physical restraints as safe	2.7 (0.8)	3.0 (2.0–3.0)	Consequences	0.55 (0.82)		
20	If I end up in a hospital, I hope staff use physical restraints on me if they deem it necessary	3.1 (1.1)	3.0 (2.0–4.0)	Consequences	0.41 (0.83)		
22	Application of physical restraints is inhumane (recoded)	3.3 (0.9)	3.0 (3.0–4.0)	Consequences	0.61 (0.81)		

df = degree of freedom; KMO = Kaiser–Meyer–Olkin criterion, α = Cronbach’s alpha, 95% CI = 95% confidence interval; SD = standard deviation; IQR = interquartile range; F1 = Consequences of restraint use for the patient; F2 = Reasons for restraint use; F3 = Appropriateness of restraint use; grey background = allocation factor.

**Table 4 ijerph-19-07144-t004:** Descriptive and factor analysis for the Discomfort and Restrictiveness scales.

		Discomfort Bartlett’s χ^2^ = 402.71, df = 13, *p*-Value < 0.000 KMO 0.77 α Full Scale (95% CI): 0.78 (0.74–0.83) Explained Variance 38% (F1D 19%; F2D 20%)				Restrictiveness Bartlett’s χ^2^ = 386.46, df = 9, *p*-Value < 0.000 KMO 0.66 α Full Scale (95% CI): 0.65 (0.58–0.72) Explained Variance 35% (F1R 19%; F2R 16%)			
Item nr.	Label	MW (SD)	Median (IQR)	F1D (α 0.90 [95% CI 0.88–0.93])	F2D (α 0.78 [95% CI 0.73–0.83])	MW (SD)	Median (IQR)	F1R (α 0.66 [95% CI 0.58–0.74])	F2R (α 0.63 [95% CI 0.54–0.71])
				factor loading (α if item is dropped)				factor loading (α if item is dropped)	
10	Wrist belt	2.9 (0.3)	3.0 (3.0–3.0)	0.89 (0.85)		2.0 (0.2)	2.0 (2.0–2.0)	0.99 (0.49)	
13	Abdominal Belt in bed	**3.0 (0.3)**	**3.0 (3.0–3.0)**	0.88 (0.86)		1.9 (0.2)	2.0 (2.0–2.0)	0.65 (0.51)	
16	Ankle belt	2.9 (0.3)	3.0 (3.0–3.0)	0.86 (0.87)		**2.9 (0.3)**	**3.0 (3.0–3.0)**	0.50 (0.66)	
08	Special sheet (fitted sheet including a coat enclosing the mattress)	2.7 (0.5)	3.0 (3.0–3.0)		0.47 (0.76)	2.8 (0.4)	3.0 (3.0–3.0)	0.34 (0.73)	
01	Sensor alarm (in bed/chair, on the floor)	**1.2 (0.4)**	**1.0 (1.0–1.0)**		0.40 (0.78)	**1.4 (0.5)**	**1.0 (1.0–2.0)**		0.34 (0.63)
02	(Wheel)Chair with table	1.8 (0.6)	2.0 (1.0–2.0)		0.66 (0.75)	2.3 (0.6)	2.0 (2.0–3.0)		0.77 (0.49)
03	Tensioning system in (wheel)chair	2.0 (0.7)	2.0 (2.0–2.0)		0.62 (0.75)	2.2 (0.6)	2.0 (2.0–3.0)		0.56 (0.57)
04	Bilateral bedrails	1.9 (0.6)	2.0 (1.0–2.0)		0.62 (0.75)	2.3 (0.5)	2.0 (2.0–3.0)		0.39 (0.58)
05	Unilateral bedrail	1.3 (0.5)	1.0 (1.0–2.0)		0.55 (0.76)	1.5 (0.5)	2.0 (1.0–2.0)		0.36 (0.61)
12	Abdominal Belt in (wheel)chair	2.6 (0.5)	3.0 (2.0–3.0)		0.40 (0.77)	1.6 (0.5)	2.0 (1.0–2.0)		0.45 (0.60)
06	Deep (wheel)chair (Siesta)	1.9 (0.7)	2.0 (1.0–2.0)		0.50 (0.76)	1.9 (0.6)	2.0 (2.0–2.0)		
07	Surveillance system	1.5 (0.6)	1.0 (1.0–2.0)		0.41 (0.77)	1.7 (0.7)	2.0 (1.0–2.0)		
09	Sleep suit (clothing that deters a person from self-undressing)	2.1 (0.7)	2.0 (2.0–3.0)		0.45 (0.77)	2.1 (0.7)	2.0 (2.0–3.0)		
11	Tightly tucked sheet (over belly and upper legs)	2.8 (0.5)	3.0 (3.0–3.0)	0.45	0.30 (0.77)	1.8 (0.4)	2.0 (2.0–2.0)		
14	Bedroom door locked	2.7 (0.6)	3.0 (3.0–3.0)			2.7 (0.5)	3.0 (2.0–3.0)		
15	Ward door locked	1.9 (0.7)	2.0 (1.0–2.0)			2.0 (0.6)	2.0 (2.0–2.0)		

df = degree of freedom; KMO = Kaiser–Meyer–Olkin criterion, α = Cronbach’s alpha, 95% CI = 95% confidence interval; SD = standard deviation; IQR = interquartile range; F1D = Fixation belts in bed; F2D = Mechanical and electronic restraint except fixation belts; F1R = Restraining the patient to the bed; F2R = Safety measures in the chair when leaving the bed or place; bold = highest/lowest descriptive scores; grey background = allocation factor.

## Data Availability

The data presented in this study are available from the corresponding author upon reasonable request. The data are not publicly available due to ethical and privacy restrictions.

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
