# Peer review of "Attitudes of Nursing Staff in Hospitals towards Restraint Use: A Cross-Sectional Study"

_ijerph, 2022, doi:10.3390/ijerph19127144_

Round 1

Reviewer 1 Report

The article is well structured, all relevant elements are adeguately presented; also some limits are declared but well discuss

Author Response

We would like to thank Reviewer 1 for the positive feedback.

Reviewer 2 Report

Dear Authors

This study has studied the attitudes of nursing staff in hospitals towards restraints use.

Although the necessary of this study is well described in the introduction part, the Methods have major problems.

  1. Study population-The authors have suggested the evidence of the sample size as 5-10 participants per item and collected 180 questionnaires. What is the total items of the questionnaire for the independent variables(attitudes, discomfort, restrictiveness).
  2. In L83-109, it seems that the original attitudes scale have 22 items,  discomfort scale 16 items, restrictiveness scale 16 items. What is the final items for each independent variables(attitudes, discomfort, restrictiveness)? Are the final validated items in the scale for attitudes, discomfort, restrictiveness are 19, 14, 10, respectively?
  3. If so, are the mean level of attitudes, discomfort, restrictiveness were suggested with the final items in table1?
  4. The cumulative % of 37% is too low. It has to be over 60%. I would rather recommend to authors to analyze with using Eigen value over 1 instead of using factor number.  Because it doesn't have to fit the subscale.
  5. The number of participants used for analyzing attitudes, discomfort, restrictiveness differs. There are ways to make up missing data. I recommend to analyzed with 182 participants. 
  6. Please shorten sentences from L149-164 and 165-188 and suggest how many items were used in factor analysis.
  7. In table 1, range of maximum and minimum range is needed instead of Median.  Did you suggest median because it did not show normal range? 

Reviewer 3 Report

The study brings insights from a nursing professional's perspective about restraint use in practice. An exciting piece of work with an excellent elaborated methodology & background, the results are similar to already conducted studies in other contexts. The response rate is higher than 50% per cent, which is a valuable fact for this type of study. 

The only critical comment is whether this study brings a certain novelty to the scientific community because there are studies already conducted on this topic.

Author Response

We would like to thank Reviewer 3 for the positive feedback. Indeed, there is already research on this topic, but in somatic acute hospitals so far mostly in intensive care units (ICU). However, as it is known that restraints are also used outside the ICU, it seems important that knowledge is generated and validated in this area so that the use of restraints can be further reduced in all health care settings.

Round 2

Reviewer 2 Report

1. Since the factor analysis was done just to show the construct validity of the tool, the reviewer ask the authors to remove table 3, table 4, result 3.2 (or suggest as Appendix). Please shorten the result 3.2 and put between L128 and L129. 

2. If the authors intention is not to develop and confirm the instrumental tool, pleases shorten 2.4 data analysis. By explaining CFA and EFA in detail, it remarks the low variance.  

3. The aim of the study is to assess nursing staffs' attitude toward restraint use(L70). But Table 2 suggest the effect of participants' characteristics toward attitude, discomfort, and restrictiveness.

->1) Please change the title of table 2.

2) Please show the correlation between attitude, discomfort, and restrictiveness

3) The reviewer suggest to input the covariance, discomfort, and restrictiveness as independent variable and attitude as dependent variable. Then the explain % would go more higher.
